# RareFlow: Physics-Aware Flow-Matching for Cross-Sensor Super-Resolution of Rare-Earth Features

## Abstract

Super-resolution (SR) for remote sensing imagery often fails under out-of-distribution (OOD) conditions, such as rare geomorphic features captured by diverse sensors, producing visually plausible but physically inaccurate results. We present RareFlow, a physics-aware SR framework designed for OOD robustness. RareFlow's core is a dual-conditioning architecture. A Gated ControlNet preserves fine-grained geometric fidelity from the low-resolution input, while textual prompts provide semantic guidance for synthesizing complex features. To ensure physically sound outputs, we introduce a multifaceted loss function that enforces both spectral and radiometric consistency with sensor properties. Furthermore, the framework quantifies its own predictive uncertainty by employing a stochastic forward pass approach; the resulting output variance directly identifies unfamiliar inputs, mitigating feature hallucination. We validate RareFlow on a new, curated benchmark of multi-sensor satellite imagery. In blind evaluations, geophysical experts rated our model's outputs as approaching the fidelity of ground truth imagery, significantly outperforming state-of-the-art baselines. This qualitative superiority is corroborated by quantitative gains in perceptual metrics, including a nearly 40% reduction in FID. RareFlow provides a robust framework for high-fidelity synthesis in data-scarce scientific domains and offers a new paradigm for controlled generation under severe domain shift.

## 1 Introduction

Monitoring rapid, small-scale environmental change requires imagery that is both high spatial resolution (for morphology) and high temporal frequency (for dynamics) Qi et al. (2025); Vu et al. (2025). Public constellations such as Sentinel-2 deliver near-global coverage with 10 m pixels and 5-day revisit, but lack the fine spatial detail needed to resolve many geomorphic features; very-high-resolution (VHR) commercial sensors provide sub-meter detail at lower cadence and are often expensive. European Space Agency (ESA) (a;b) This mismatch creates a persistent spatiotemporal gap for near-real-time environmental monitoring. Cross-sensor super-resolution (SR) has significant scientific value, as it promises to synthesize VHR content from publicly available lower-resolution satellite imagery, substantially reducing reliance on commercial satellite data. However, little SR research explicitly tackles this problem or reliably addresses the accompanying distribution-shift challenge. Qi et al. (2025)

Diffusion-based SR now dominates perceptual-quality benchmarks and offers precise structural control via conditional mechanisms (e.g., ControlNet Zhang & Agrawala (2023)), yet its success typically hinges on two assumptions that fail in scientific RS: (i) the low-resolution (LR) input is a structurally faithful proxy for the desired high-resolution (HR) scene; (ii) the model's prior has seen enough semantically similar examples to render the target phenomenon. When either assumption breaks—blurred structure or out-of-distribution (OOD) semantics—generators can hallucinate textures that look plausible but violate physics and radiometry. This tension echoes the perception–distortion trade-off: pushing perceptual realism often degrades fidelity, and vice versa. Moser et al. (2024); Saharia et al. (2021); Blau & Michaeli (2018)

In remote sensing (RS), the primary objective is scientific fidelity rather than photorealistic reconstruction Wang et al. (2022b). First, generative models must preserve precise spectral signatures, a constraint that generic models routinely violate, invalidating downstream analysis. Dong et al. (2021); Dou et al. (2020) Second, the ignorant prior problem becomes a critical point of failure. A model cannot generate a geologically sound rare-earth feature from aesthetic principles alone; it requires an implicit understanding of geomorphology. Lacking this, it produces plausible-looking artifacts, not scientifically valid image. This problem is intractable for standard methods, as the very rarity of the phenomena under study makes it impossible to amass the vast datasets these models typically require. Chen et al. (2024); Liu et al. (2022)

These challenges are magnified for rare landforms, such as retrogressive thaw slumps (RTS), ice-rich permafrost failures whose occurrence and expansion have intensified under Arctic warming. Their rarity yields extreme class imbalance, severe few-shot regimes, and strong OOD shift across sensors and regions. Any SR framework used for RTS must therefore (i) recognize OOD conditions and reduce prior creativity when evidence is weak, and (ii) ground synthesis in physics, not aesthetics. Nesterova et al. (2024); Lewkowicz & Way (2019); Barth et al. (2025); Nitze et al. (2025)

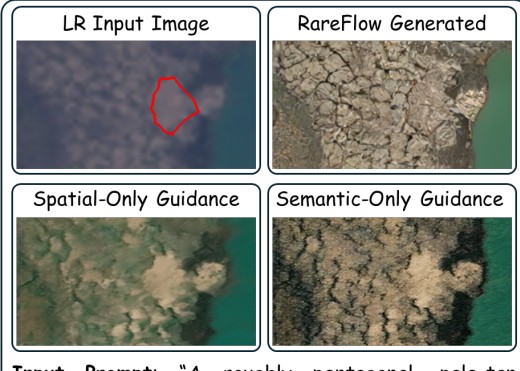

**Input Prompt**: "A roughly pentagonal, pale-tan retrogressive thaw slump with exposed bare-soil headscarp and a lobate toe cuts into patterned ice-wedge polygon tundra and abuts the bright turquoise shoreline of a small lake."

Figure 1: When the LR input is blurry or semantically OOD, spatial-only guidance preserves coarse morphology yet remains soft, while semantic-only guidance hallucinates plausible—but incorrect—textures. *RareFlow* balances structural evidence from the LR image with textual semantics, suppressing hallucinations and preserving physically consistent geometry and spectra.

Confronting this triad of challenges—blurry guidance, absent priors, and data scarcity—demands more than incremental improvements; it requires a framework explicitly designed for this conflict. We therefore propose RareFlow, a physics-aware SR framework tailored for cross-sensor RS under severe domain shift, designed for OOD robustness. Our work reveal a fundamental tension as illustrated in Figure 1: strong spatial conditioning alone preserves coarse structure but also propagates undesirable blur, while semantic guidance alone generates realistic features but sacrifices geometric fidelity. RareFlow is explicitly designed to resolve this conflict. Its dual-conditioning mechanism uses a gated ControlNet (Zhang & Agrawala, 2023) to dynamically weigh the blurry source image for structural integrity while simultaneously leveraging textual prompts to provide the rich semantic context required to synthesize scientifically plausible details, even when the target phenomenon is rare and visually complex.

**Contributions.** The main contributions of this paper are summarized as follows.

1. **Dynamic Control of Priors to Mitigate Hallucination.** We introduce a dual-conditioning framework where a Gated ControlNet preserves geometric fidelity from the LR input, while text prompts guide semantic synthesis. This gating mechanism dynamically balances the two influences, reducing reliance on learned priors for OOD data and preventing feature hallucination.

2. **A Physics-Aware Loss for Preserving Scientific Imagery.** We design a multifaceted loss function that anchors the model's output to the ground truth. By enforcing consistency in both the spectral domain and a perceptually uniform color space, we ensure the SR process preserves the large-scale radiometric information captured by the sensor and prevents the hallucination of erroneous detail.

3. **Unified Framework for Harmonization and SR.** We present a single, end-to-end model that jointly performs SR and radiometric harmonization. This removes the need for separate

pre-processing pipelines and ensures color and brightness statistics are consistent with a target reference during the reconstruction process.

4. **State-of-the-Art (SOTA) Performance in a Low-Data Regime.** On a curated multi-sensor earth-observation benchmark emphasizing RTS and with only $\approx 800$ labeled RTS images, RareFlow reconstructs fine-scale structures and surpasses strong baselines confirmed by both quantitative metrics and qualitative analysis; an expert study further evaluated superior fidelity and scientific integrity.

## 2 RELATED WORKS

Traditional SR methods, from classical interpolation to early deep learning approaches, often struggled to reconstruct fine, high-frequency details Johnson et al. (2016); Ledig et al. (2017); Sajjadi et al. (2017); Blau & Michaeli (2018); Moser et al. (2024). Their limitation tends to average out plausible solutions and results in overly smooth or blurry textures Blau & Michaeli (2018); Johnson et al. (2016).

Diffusion models address this by learning the distribution of natural images and sampling from it, enabling plausible, high-fidelity detail beyond simple sharpening Ho et al. (2020); Song et al. (2021); Saharia et al. (2023); Moser et al. (2024). This generative capability underlies SOTA perceptual SR, but applying it to remote sensing demands (i) strict geometric fidelity, (ii) scientific/radiometric plausibility, and (iii) robustness across sensors and styles Lanaras et al. (2018b); Gascon et al. (2017); Scarpa & Ciotola (2022a); Claverie et al. (2018); Ju et al. (2025). Our work is situated at the intersection of these three key challenges.

**Enforcing Geometric and Structural Fidelity** A primary challenge in SR is maintaining strict fidelity to the geometric and structural content of the LR input. A prominent line of work injects strong *spatial conditioning*: ControlNet-centric designs like ControlSR Zhang et al. (2023); Wan et al. (2024) enhance geometric faithfulness by injecting LR spatial cues at multiple scales. Other methods leverage segmentation masks or edge priors including SAM-DiffSR Wang et al. (2024) and SAMSR Liu et al. (2025), to enforce sharp boundaries and object consistency. A second strategy involves using *structure-aware objectives* that directly penalize geometric distortion during training. SPSR restores and supervises image gradients, using a gradient branch and gradient loss to preserve edges and linework Ma et al. (2020). To avoid hallucinations that violate the LR measurement, *data-consistent diffusion* methods impose the forward model during sampling: DDRM and DDNM enforce measurement consistency for SR and other linear inverse problems, while hard data-consistency with latent diffusion (ReSample) projects samples back to the measurement manifold at each step Kawar et al. (2022); Wang et al. (2022c); Song et al. (2024).

Finally, *physics-aware models* explicitly embed knowledge of the image acquisition process. These methods often incorporate the sensor's Point Spread Function (PSF) or noise characteristics into a forward degradation model, ensuring the output is physically plausible Yang & Ren (2010). In remote sensing, this is extended by using guidance from other spectral bands or sensor physics to preserve features like building footprints Lanaras et al. (2018a); Armannsson et al. (2021); Scarpa & Ciotola (2022b). While effective for known degradations, these approaches primarily constrain the degradation process rather than the physical properties (e.g., spectral profiles) of the final HR output itself.

Nevertheless, a key limitation of these approaches is their reliance on robust guidance signals like accurate segmentation maps. For rare and subtly-defined geomorphological features like RTS, accurate segmentation masks are prohibitively expensive to create and are often brittle in practice. These features lack the clear, consistent boundaries that segmentation-heavy models rely on. This limitation motivates our use of a more flexible gated ControlNet, which provides robust structural guidance directly from the LR image without depending on fragile, external annotations.

**Ensuring Semantic Plausibility for Scientific Use** Beyond structural accuracy, SR outputs must be *semantically* and *scientifically* correct; plausible is not the same as correct. Recent work improves faithfulness by stabilizing the denoising trajectory and balancing LR evidence with generative priors: timestep-aware fusion preserves input semantics early while enabling detail later Lin et al. (2024), and joint fine-tuning with alignment/consistency objectives reduces hallucination and better anchors content to the LR input Chen et al. (2025b). A complementary direction injects richer se-

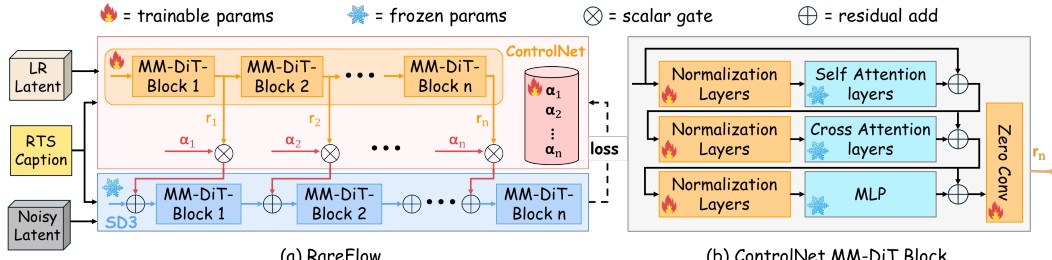

Figure 2: **(a) The training architecture of RareFlow**. The control path (orange) consumes LR latents and caption tokens to produce residual hints $r_i$ and predicts per-block scalars $\alpha^l \in [0,1]$ that scale $r_i$ before injection into the frozen backbone (blue) features $F_i \leftarrow F_i + \alpha_i r_i$. **(b)** ControlNet MM-DiT block internals which produces $\alpha_i$ (see Sec. 3.1).

mantic priors so the model "knows what it is restoring": cross-modal guidance via MLLMs provides scene- and object-level cues Qu et al. (2024), while segmentation-conditioned diffusion explicitly constrains regions to class-consistent appearance Xiao et al. (2024); this builds on earlier semantic SR showing that category-aware conditioning reduces implausible textures Wang et al. (2018).

However, these methods are optimized for generic photo domains. They excel at producing plausible textures for common objects but lack the domain-specific knowledge to reconstruct the unique morphology of a thaw slump or distinguish it from similar-looking terrain. Their reliance on general-world semantics risks generating photorealistic yet scientifically inaccurate artifacts. Our work directly addresses this gap by using highly specific textual prompts to encode geological knowledge, guiding the model to generate features that are not just visually convincing but also geologically plausible.

**Cross-Sensor Style Transfer in SR** A critical challenge in satellite imaging, often overlooked by general-purpose SR models, is the pronounced stylistic variation across sensors. Each platform exhibits distinct radiometry, spectral bandpasses, point-spread/MTF characteristics, noise statistics, and sun–sensor geometries, producing domain shifts that alter texture and color distributions Gascon et al. (2017); European Space Agency (ESA) (c); Claverie et al. (2018); Ju et al. (2025). Most multi-sensor SR and fusion methods—including recent diffusion-based pipelines such as DiffFuSR and SatDiffMoE—treat these differences as a nuisance to be harmonized or fused into a canonical representation, rather than reproducing a target sensor's style Sarmad et al. (2025); Luo et al. (2024). In contrast, cross-sensor domain-adaptation studies in recognition consistently document substantial performance gaps attributable to sensor/style shift, underscoring that the discrepancy is persistent and non-trivial Wang et al. (2022a); Li et al. (2025); Zeng et al. (2024). Consequently, explicit *style transfer* across sensors within SR—achieving a target instrument's textural and radiometric signature while preserving geometry—remains underexplored; doing so requires sensor-aware conditioning and physics-aware forward models (e.g., bandpass and MTF matching), together with evaluations that assess both radiometric agreement and style fidelity beyond generic distortion metrics Claverie et al. (2018); Ju et al. (2025); Scarpa & Ciotola (2022a).

In contrast, we treat sensor style as a valuable signal. We reframe the problem not as style harmonization, but as explicit style transfer within the SR task. This novel perspective allows our model to learn the specific visual signature of a target sensor (e.g., higher contrast, unique textural patterns) and intelligently apply it to an image from a different source. This reframing provides fine-grained control over the final appearance, enabling both detail enhancement and stylistic consistency with the characteristics of the chosen sensor. Our work explicitly tackles this dual objective for rare geological features, filling a crucial gap in the remote sensing literature.

## 3 RareFlow: A Physics-Aware Dual-Conditioned Flow Matching

We work in latent space with a frozen VAE $(\mathcal{E}, \mathcal{D})$ and a frozen *diffusion transformer* $f_\theta$ (SD3 MMDiT) Esser et al. (2024); Peebles & Xie (2023). Given an HR image $y$, we encode $z_0 = \mathcal{E}(y)$ and build noisy latents via the SD3 flow-match schedule

$$z_t = (1 - \sigma_t) z_0 + \sigma_t \epsilon, \qquad \epsilon \sim \mathcal{N}(0, \mathbf{I}), \tag{1}$$

where $\{\sigma_t\}$ is the discrete schedule of the FlowMatch Euler solver. The denoiser receives $(z_t, t)$ together with text conditioning and ControlNet residuals.

## 3.1 CONTROL ADAPTERS WITH UNCERTAINTY-GATED SCALARS

A ControlNet $g_\phi$ processes the LR *latent* $\tilde{x} = \mathcal{E}(x)$ and emits per-block residual feature maps $\{r^l\}_{l \in \mathcal{L}}$ aligned with the transformer blocks Zhang & Agrawala (2023); Zavadski et al. (2024). To modulate these residuals in a *shape-preserving* way, we learn a scalar gate per block that depends on diffusion time and uncertainty:

$$\alpha^l(t, u) = \sigma\Big(p_0^l + p_t^l \cdot \text{norm}(t) + p_u^l \cdot u\Big), \tag{2}$$

$$\tilde{r}^l = s_{\text{ctrl}} \, \alpha^l(t, u) \, r^l, \qquad s_{\text{ctrl}} \in \mathbb{R}_+, \tag{3}$$

where $\sigma(\cdot)$ is the logistic function and $\text{norm}(t) \in [0, 1]$ is a normalized time index. The scalar $u \in [0, 1]$ summarizes *only* MC-dropout uncertainty.

## 3.2 UNCERTAINTY VIA MONTE CARLO DROPOUT

We estimate epistemic uncertainty with MC dropout Gal & Ghahramani (2016) while freezing all backbones. Dropout (rate $p_{\text{do}}$) is enabled only in the trainable ControlNet $g_\phi$ and $\alpha^l(\cdot)$. For a fixed $(x, c)$ we draw $T$ stochastic reconstructions

$$\hat{y}^{(k)} = \mathcal{D}\Big(f_\theta\big(z_t, t, c, x; \; g_\phi \text{ with dropout}\big)\Big), \quad k = 1, \dots, T,$$

$$v(i, j) = \text{Var}_{k=1}^T\big[\hat{y}_{i,j}^{(k)}\big], \tag{4}$$

$$u = \text{clip}\Big(\tfrac{\text{mean}(\text{clip}(v, 0, \tau))}{\tau}, \, 0, \, 1\Big).$$

We set the scale on-the-fly as

$$\tau = \text{Pct}_{95}\big(\{\text{mean}(v)\}_{\text{mini-batch}}\big) + \epsilon, \tag{5}$$

or equivalently use a fixed scale $\kappa > 0$ via

$$u = 1 - \exp\Big(-\tfrac{\text{mean}(v)}{\kappa}\Big). \tag{6}$$

During sampling, $u$ modulates the gates via Eq. equation 2.

## 3.3 PHYSICS-AWARE TRAINING OBJECTIVE

We train the control pathway $g_\phi$ and gate parameters $\{p_0^l, p_t^l, p_u^l\}$ while keeping $(\mathcal{E}, \mathcal{D})$ and $f_\theta$ frozen. For training pairs $(x, y)$, define

$$\hat{y} = \mathcal{D}(f_\theta(z_t, t, c, x)), \tag{7}$$

and the preconditioned FlowMatch loss Lipman et al. (2022)

$$\mathcal{L}_{\text{base}} = \mathbb{E}_{(x,y), \, t, \, \epsilon}\Big[\big\|\omega(\sigma_t)\left(f_\theta(z_t, t, c, x) - z_0\right)\big\|_2^2\Big], \tag{8}$$

with schedule-specific weight $\omega(\sigma_t)$. Although effective, this base term alone can produce overly smooth outputs; we therefore add three complementary losses.

**Frequency alignment (spectral magnitude).** To discourage oversmoothing and limit hallucinated detail, we align spectral magnitudes in either latent or pixel space with a radial emphasis on mid/high frequencies Jiang et al. (2021); Fuoli et al. (2021):

$$\mathcal{L}_{\text{fft}} = \Big\| W \odot \Big(\big|\mathcal{F}(u_\theta)\big| - \big|\mathcal{F}(u^\star)\big|\Big) \Big\|_1, \tag{9}$$

where $u_\theta$ and $u^\star$ denote latents, and $W(\rho) \propto \rho^\gamma$ stresses mid/high spatial frequencies that SR tends to lose.

**Radiometric Consistency (perceptual color).** To preserve coarse color/brightness, we compare in CIELAB after blur. Let $\text{Lab}(u) \in \mathbb{R}^{H \times W \times 3}$, $G_b$ be channelwise Gaussian blur, and $\text{Lab}_b(\cdot) \equiv G_b(\text{Lab}(\cdot))$. With per-channel spatial mean $\mu(\cdot)$ and stdev $\sigma(\cdot)$,

$$\mu(u) = \text{spatial mean per channel}, \qquad \sigma(u) = \text{spatial stdev per channel}, \tag{10}$$

$$\mathcal{L}_{\text{color}} = \big\| \text{Lab}_b(\hat{y}) - \text{Lab}_b(y) \big\|_1 + \sum_{i \in \{\mu, \sigma\}} \big\| i\big(\text{Lab}_b(\hat{y})\big) - i\big(\text{Lab}_b(y)\big) \big\|_1. \tag{11}$$

**Total objective.** Finally, we add LPIPS (AlexNet) Zhang et al. (2018a) to better correlate with human judgment. Our physics-aware loss enforces spectral and perceptual color consistency, preserving large-scale radiometry while discouraging hallucinated detail:

$$\mathcal{L} = \mathcal{L}_{\text{base}} + \lambda_{\text{fft}} \mathcal{L}_{\text{fft}} + \lambda_{\text{color}} \mathcal{L}_{\text{color}} + \lambda_{\text{lpips}} \mathcal{L}_{\text{lpips}}. \tag{12}$$

## 4 EXPERIMENTS AND ANALYSIS

We validate our approach on a challenging, real-world dataset for remote sensing SR. The task is to learn a mapping from 10m Sentinel-2 (LR) images to 2m Maxar (HR) ground truth. This cross-sensor and cross-temporal setup inherently induces OOD conditions and presents several significant challenges, illustrated in Fig. 3.

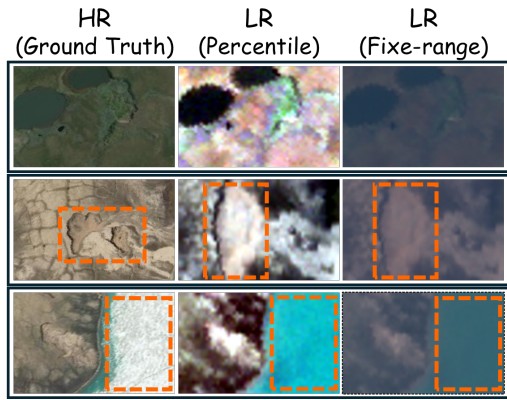

These challenges include: **(1) Spatio-temporal misalignment** between LR and HR images, acquired at different times, causes sub-pixel shifts, dramatic variations in illumination, and stark land cover changes. Furthermore, the dataset is characterized by **(2) Small image dimensions**, with inputs as small as 30×40 pixels, which prevent direct comparison to models evaluated on larger benchmark images. **(3) A non-standard 12-bit data range** that departs from the typical 8-bit format and makes model performance highly sensitive to the chosen normalization method as it materially alters input data distribution and model performance. **(4) A limited training corpus of $\approx$ 800 images**, which necessitates a data-efficient approach unsuitable for training large models from scratch. Further details are provided in Appendix .1.

Figure 3: Data challenges. Left to Right: HR (Maxar), LR (Sentinel-2, Percentile Norm), LR (Sentinel-2, Fixed Norm). Row 1 shows color discrepancies and the effect of normalization. Row 2 shows spatial misalignment. Row 3 shows temporal misalignment due to changes in snow cover.

### 4.1 EVALUATION STRATEGY

We compare our method against SOTA SR models from two categories. First, we include architectures specifically tailored for remote sensing and trained on Sentinel-2 LR images, such as OpenSR (Donike et al., 2025), MISR-S2 (Okabayashi et al.) and ZoomLDM (Yellapragada et al., 2025). Second, we benchmark against leading general-purpose methods like AdcSr (Chen et al., 2025a), SamSR (Liu et al., 2025) and SeeSR (Wu et al., 2024) to test whether their powerful architectures offer a fundamental advantage that translates to the remote sensing domain.

Our evaluation strategy involves testing in two settings. We first benchmark all models on the real-world, cross-sensor pairs. We then use an idealized setting with synthetically downsampled HR images to isolate architectural performance. The quantitative metrics used for these evaluations are described and justified in detail in Appendix .2.

### 4.2 QUANTITATIVE ANALYSIS: THE FIDELITY VS. REALISM TRADE-OFF

**Quantitative trends.** Table 1 summarizes fidelity (PSNR/SSIM/FSIM), perceptual similarity (LPIPS/DISTS), and no-reference realism (FID/NIQE/MANIQA). RareFlow attains the lowest

Figure 4: Qualitative comparison on paired LR–HR data.

LPIPS (0.36) and DISTS (0.30), and the lowest FID (116.16). Relative to the next-best FID (AdcSR, 187.18), this is a 38% reduction. On fidelity, RareFlow achieves the best SSIM (0.59) and FSIM (0.83) while remaining competitive in PSNR (18.76 dB vs. 18.78 dB for SeeSR;) These results are consistent with the well-known perception–distortion trade-off (Blau & Michaeli, 2018): methods optimized primarily for PSNR tend to underperform on perceptual/realism metrics, whereas RareFlow improves perceptual quality without materially degrading pixelwise fidelity.

Table 1: SR results on **paired LR–HR** data. **Bold** = best, underline = second-best; arrows indicate whether higher or lower is better.

| SR Model | Fidelity Metrics | | | Perceptual Similarity | | Realism Metrics | | |
|---|---|---|---|---|---|---|---|---|
| | PSNR ↑ | SSIM ↑ | FSIM ↑ | LPIPS ↓ | DISTS ↓ | FID ↓ | NIQE ↓ | MANIQA ↑ |
| ZoomLDM | 17.23 | 0.26 | 0.47 | 0.60 | 0.59 | 352.11 | 18.10 | 0.19 |
| SeeSR | **18.78** | 0.50 | 0.71 | 0.46 | 0.38 | 302.36 | 10.78 | **0.36** |
| AdcSR | 18.59 | 0.58 | 0.71 | 0.40 | 0.37 | 187.18 | 8.38 | 0.28 |
| MISR-S2 | 18.39 | 0.50 | 0.68 | 0.54 | 0.43 | 254.70 | 13.55 | 0.33 |
| SAMSR | 18.36 | 0.54 | 0.74 | 0.48 | 0.39 | 189.01 | 11.84 | 0.32 |
| OpenSR | 17.29 | 0.51 | 0.66 | 0.41 | 0.36 | 225.62 | 9.80 | 0.25 |
| RareFlow (Ours) | 18.76 | **0.59** | **0.83** | **0.36** | **0.30** | **116.16** | **5.36** | 0.31 |

Beyond pure resolution enhancement, the core challenge of this task is performing simultaneous SR and cross-sensor style transfer. A successful model must bridge the significant domain gap between the Sentinel-2 source and the Maxar target. Figure 4 provides a striking visual demonstration of this challenge, illustrating the fundamental limitations of prior methods and the scientific utility of our approach. The baseline models fundamentally fail at the style transfer aspect. As shown in the figure, their outputs largely retain the characteristic radiometric properties of the Sentinel-2 input, such as muted contrast and a smoother textural profile. Consequently, the baselines' attempts at SR only sharpen the wrong stylistic representation. RareFlow, however, excels at this joint task by simultaneously reconstructing geological details and translating the image into the target Maxar style, producing an output that is both HR and stylistically faithful to the ground truth.

### 4.2.1 CONTROLLED SR (HR→LR DOWNSAMPLING)

To isolate the SR capacity from the cross-sensor harmonization, we synthetically downsample HR Maxar and evaluate 3-channel RGB methods under an SR protocol (details in App. .4). We exclude methods that rely on non-RGB bands, pan-sharpening inputs, auxiliary geospatial metadata, or sensor-specific priors, to ensure an apples-to-apples comparison with our RGB-only pipeline.

On HR-downsampled-HR pairs as shown in Table 2, RareFlow delivers a clearly better distortion–perception trade-off: vs. the strongest fidelity-oriented baseline (SeeSR), it cuts perceptual/realism error by $-27.8\%$ LPIPS, $-21.7\%$ DISTS, $-41.7\%$ FID, and $-31.0\%$ NIQE, while staying within 4% PSNR (SSIM ties; FSIM 1.2%). Against the closest realism challenger (SAMSR). Qualitative comparisons in the appendix .4 corroborate these trends on HR-downsampled→HR inputs, where RareFlow reconstructs sharper textures across diverse scenes.

This experiment confirms that the core components of our architecture provide a powerful and generalizable foundation for image restoration. The leading performance in both the specialized cross-

Table 2: SR results on **HR-downsampled–HR pairs** data. **Bold** = best, underline = second-best; arrows indicate whether higher or lower is better.

| SR Model | Fidelity Metrics | | | Perceptual Similarity | | Realism Metrics | | |
|---|---|---|---|---|---|---|---|---|
| | PSNR ↑ | SSIM ↑ | FSIM ↑ | LPIPS ↓ | DISTS ↓ | FID ↓ | NIQE ↓ | MANIQA ↑ |
| AdcSR | 26.59 | 0.65 | 0.80 | 0.27 | 0.29 | 191.50 | 5.91 | 0.28 |
| SeeSR | **29.37** | **0.75** | 0.86 | 0.18 | 0.23 | 141.48 | 6.26 | 0.28 |
| ZoomLDM | 21.60 | 0.54 | 0.66 | 0.47 | 0.51 | 232.80 | 8.10 | 0.11 |
| SAMSR | 27.59 | 0.65 | 0.82 | 0.22 | 0.23 | 128.20 | 5.20 | 0.22 |
| RareFlow (Ours) | 28.20 | **0.75** | **0.87** | **0.13** | **0.18** | **82.53** | **4.32** | **0.46** |

sensor task and the standard SISR task strongly suggests that our model's design is robust and highly effective, making a significant contribution to the broader field of SR.

### 4.3 HUMAN EVALUATION BY DOMAIN EXPERTS

To validate the scientific utility of our results, we conducted a two-stage evaluation with geomorphology experts (details in Appendix .5).

**Stage 1: Validating Semantic Guidance Integrity.** Given a HR reference image with a binary mask over the region of interest, we prompt a SOTA vision–language model (GPT-5) OpenAI (2025) to produce a structured, resolution-aware description of RTS-predictive attributes (feature texture, shape, and immediate environment), see Appendix .3 for prompt design. This foundational check ensures our model is conditioned on scientifically sound semantic priors.

**Stage 2: Evaluating Super-Resolved Imagery.** With the integrity of the language guidance established, we proceeded to the main evaluation of our model's SR capabilities, again leveraging domain experts. The study was designed to answer two key questions: whether our SR images offered a clarity improvement over the LR inputs, and the more challenging test of whether they could achieve perceptual parity with the HR ground truth. The results were conclusive. Experts unanimously agreed that RareFlow's outputs provide a consistent and significant clarity enhancement over the blurry LR inputs. More critically, in numerous instances, our model achieved the gold standard: experts judged the super-resolved images to be perceptually on par with the 2m Maxar ground truth, successfully reconstructing fine-scale RTS features that were ambiguous or invisible in the source image.

### 4.4 ABLATION STUDIES: DISSECTING THE SOURCE OF PERFORMANCE

We performed comprehensive ablations (Table 3) to isolate the contributions of our key components: spatial conditioning (ControlNet), semantic guidance (text captions), and perceptual losses. Our analysis uncovers a critical dilemma inherent to this task.

We conduct a comprehensive ablation across 6 configurations to isolate the effects of (i) *spatial priors* (ControlNet: pre-trained vs. scratch, and an $\alpha$-gated conditioning strength), (ii) *semantic guidance* (caption supervision describing RTS content), and (iii) *perceptual regularization* (FFT, color consistency, and LPIPS losses). The task presents unique challenges arising from two principal factors: (1) ground-truth (GT) imagery is blurred and compressed; this limitation means that classic fidelity metrics (like PSNR, SSIM) can be misleading, as a high score might only reflect a model's ability to reproduce undesirable blur, and (2) RTS scenes exhibit distinctive geomorphology. Therefore, while we report standard fidelity metrics for completeness, our analysis gives greater weight to perceptual metrics (LPIPS, DISTS) and no-reference metrics (FID, NIQE, MANIQA) which better assess the generated image's realism and visual quality.

Our analysis of Table 3 highlights a clear trade-off. The model with only a pre-trained ControlNet (2) excels at most fidelity and perceptual similarity metrics. This shows it is highly effective at replicating the GT's structure, but it also reproduces its undesirable softness, leading to poor scores on no-reference realism metrics like MANIQA. In contrast, adding caption guidance (3, 5) boosts realism, achieving the best MANIQA scores and significantly lowering FID. However, this comes at a steep price, as fidelity metrics collapse indicating that the model is generating plausible but structurally incorrect content based on the descriptive captions.

Table 3: Comparison of model variants. **Bold** = best, underline = second-best; arrows indicate whether higher or lower is better.

| Model Configuration | Fidelity Metrics | | | Perceptual Similarity | | Realism Metrics | | |
|---|---|---|---|---|---|---|---|---|
| | PSNR ↑ | SSIM ↑ | FSIM ↑ | LPIPS ↓ | DISTS ↓ | FID ↓ | NIQE ↓ | MANIQA ↑ |
| (1) Baseline (from Scratch) | 18.01 | 0.49 | 0.81 | 0.41 | 0.33 | 206.32 | 5.98 | 0.25 |
| (2) + Pre-trained CN | **18.80** | 0.51 | **0.85** | **0.35** | **0.29** | 187.60 | 6.42 | 0.19 |
| (3) + Caption | 17.08 | 0.38 | 0.74 | 0.47 | 0.32 | 145.43 | 5.64 | **0.32** |
| (4) + Caption & $\alpha$-gate | 17.69 | 0.52 | 0.81 | 0.38 | 0.30 | 138.21 | 5.78 | 0.23 |
| (5) + Caption & Loss | 17.11 | 0.38 | 0.74 | 0.47 | 0.32 | 144.61 | 5.61 | **0.32** |
| **(6) Full Model (Ours)** | 18.76 | **0.59** | 0.83 | 0.36 | 0.30 | **116.16** | **5.36** | 0.31 |

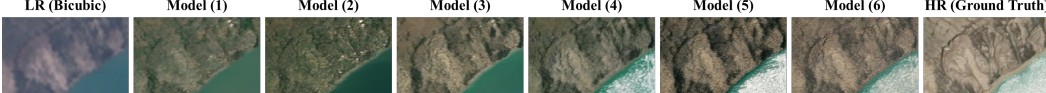

Figure 5: Visual Comparison of model variants.

RareFlow (**full model (6)**) successfully navigates this dilemma. It achieves the best distributional realism and lowest artifacts, evidenced by its SOTA FID and NIQE scores. At the same time, it maintains excellent fidelity, securing the second-best scores across all five fidelity and perceptual similarity metrics. This demonstrates that our combined approach does not sacrifice structural integrity for realism, but instead achieves a synergistic balance, yielding outputs that are both geometrically faithful and perceptually convincing.

Table 4: Ablation on dataset averages. **Bold** = best, underline = second-best; arrows indicate whether higher or lower is better.

| Model | Accuracy & Calibration | | | | Perceptual Quality | | |
|---|---|---|---|---|---|---|---|
| | SAM ↓ | $\Delta E_{2000}$ ↓ | NLL (Gauss) ↓ | ECE ↓ | QNR ↑ | $D_\lambda$ ↓ | $D_s$ ↓ |
| Model (4) | 4.5318 | 11.1550 | 7.1115 | 0.3853 | **0.4704** | **0.1036** | 0.4745 |
| Model (5) | 3.4769 | 10.3439 | 0.2251 | 0.2115 | 0.4665 | 0.1292 | **0.4676** |
| Model (6) | **2.6088** | **6.4504** | **-0.7792** | **0.1600** | 0.3312 | 0.1201 | 0.6211 |

Table 4 shows that progressively adding *captions* and then the *alpha-gated physics-aware loss* (Model 4→5→6) **monotonically improves accuracy and calibration**—SAM $4.53 \rightarrow 2.61°$, $\Delta E_{2000}$ $11.16 \rightarrow 6.45$, NLL $7.11 \rightarrow -0.78$, and ECE $0.385 \rightarrow 0.160$—with the **full model** (Model 6),: captions + alpha-gated physics) best overall; the **captions-only** variant (Model 5) outperforms the baseline (Model 4), and the additional physics term trades off perceptual quality as QNR decreases ($0.470 \rightarrow 0.331$) due to a higher $D_s$ ($\approx 0.47 \rightarrow 0.62$) while $D_\lambda$ remains low ($\approx 0.12$).

# 5 CONCLUSIONS

In this work, we address a critical failure mode of generative models: the synthesis of physically implausible details when generating from OOD scientific data in extreme low-data regimes. To mitigate this, we introduce RareFlow, a framework that maintains generative fidelity under significant domain shifts via a novel physics-aware dual-conditioning mechanism. By preserving geometric structure while gating the influence of semantic guidance, RareFlow avoids hallucinating invalid features, enabling the synthesis of plausible instances from the true data distribution rather than mere pixel replication. Our benchmark results show that RareFlow outperforms all baselines in fidelity. While this emphasis may constrain sample diversity in the most data-scarce settings, ablations reveal tunable trade-offs through gating hyperparameters. Overall, RareFlow demonstrates how to enforce scientific fidelity and cross-domain generalization in heterogeneous, low-resource settings. Future extensions of its dual-conditioning and gating principles to temporal domains, such as video synthesis, could further reduce distributional drift and hallucinations in foundation models trained on multimodal, cross-sensor inputs.

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

# APPENDIX

## .1 DATASETS & CHALLENGES

The dataset covers seven Arctic regions, including the Yamal and Gydan Peninsulas, Lena River, and Kolguev Island in Russia, along with Herschel Island, Horton Delta, Tuktoyaktuk Peninsula, and

Banks Island in Canada Yang et al. (2023). These sites provide diverse land-cover characteristics, including tundra, ice-rich permafrost bluffs, and coastal slopes, making the dataset valuable for capturing high-frequency spatial patterns in satellite imagery. The data set contains RTS annotations that were generated through careful manual digitization of Maxar imagery, supported by multi-temporal cross-checks with Sentinel-2 and ArcticDEM Yang et al. (2023).

Although the source datasets contain multiple spectral channels and auxiliary layers (e.g., NDVI, elevation), this work intentionally restricts the model's input and output to the RGB color space. This design choice is motivated by the goal of enhancing the generalizability of the framework beyond the domain of multi-spectral remote sensing. By focusing on standard RGB data, the proposed method is more readily adaptable to other fields where training data is often limited to three channels and may be scarce, such as medical imaging or archival photograph restoration. Our method address several difficulties inherent to the real-world SR data.

To handle the 12-bit radiometric resolution of the Sentinel-2 LR images, we considered two primary normalization methods to scale pixel values to the [0,1] range. Let $X \in \mathbb{R}^{H \times W \times B}$ be the input image (height $H$, width $W$, $B$ bands), and let $Y$ be the normalized output with the same shape.

### .1.1 PER-BAND PERCENTILE NORMALIZATION

For each band $b \in \{1, \ldots, B\}$, compute the lower/upper percentile values

$$\alpha_b = Q_{p_{\text{low}}}(X_{\cdot,\cdot,b}), \qquad \beta_b = Q_{p_{\text{high}}}(X_{\cdot,\cdot,b}). \tag{13}$$

Then for every pixel $(i, j)$:

$$Y_{i,j,b} = \begin{cases} 0, & \text{if } \beta_b = \alpha_b, \\ \text{clip}\left( \dfrac{X_{i,j,b} - \alpha_b}{\beta_b - \alpha_b}, 0, 1 \right), & \text{otherwise.} \end{cases} \tag{14}$$

Equivalently (vectorized per band):

$$Y_{\cdot,\cdot,b} = \text{clip}\left( \frac{X_{\cdot,\cdot,b} - \alpha_b}{\beta_b - \alpha_b}, 0, 1 \right), \quad \text{with the same degenerate case } (\beta_b = \alpha_b). \tag{15}$$

### .1.2 FIXED-RANGE NORMALIZATION

Given constants $m = \text{min\_val}$ and $M = \text{max\_val}$ (e.g., $m = 0$, $M = 3000$):

$$Y = \begin{cases} 0, & \text{if } M = m, \\ \text{clip}\left( \dfrac{X - m}{M - m}, 0, 1 \right), & \text{otherwise,} \end{cases} \tag{16}$$

applied element-wise.

**Note.** $\text{clip}(z, 0, 1) = \min\{1, \max\{0, z\}\}$ is applied element-wise, and $Q_p(\cdot)$ denotes the $p$-th percentile computed over all pixels of the given band.

## .2 EVALUATION METRICS

We evaluate SR models using a mixture of full-reference (FR), no-reference (NR), and set-level distributional metrics and we standardize implementation details to ensure fair comparison across methods. The most fundamental metric is the Peak Signal-to-Noise Ratio (PSNR) Hore & Ziou (2010). It's simple and fast, directly measuring the pixel-wise difference between a generated image and the ground truth. While useful for gauging raw reconstruction accuracy, its major weakness is its poor correlation with human vision; an image with a high PSNR can still look unnatural or blurry to our eyes. To bridge this gap between numerical error and perceived quality, the Structural Similarity Index Measure (SSIM) Wang et al. (2004) was introduced. Instead of comparing pixels in isolation, SSIM evaluates the similarity of luminance, contrast, and structure, offering a more perceptually relevant score.

**PSNR**  Given a reference image $x \in [0,R]^{H \times W}$ and a reconstruction $y$, with mean squared error

$$\text{MSE}(x,y) \; = \; \frac{1}{HW} \sum_{i=1}^{H} \sum_{j=1}^{W} \big(x_{ij} - y_{ij}\big)^2, \tag{17}$$

$$\text{PSNR}(x,y) \; = \; 10 \log_{10}\!\Big(\tfrac{R^2}{\text{MSE}(x,y)}\Big) \; [\text{dB}], \tag{18}$$

where $R$ is the peak pixel value (e.g., $R{=}255$ for 8-bit).

**SSIM**  It compares local luminance ($l$), contrast ($c$), and structure ($s$) between $x$ and $y$:

$$\text{SSIM}(x,y) \; = \; \frac{(2\mu_x\mu_y + C_1)(2\sigma_{xy} + C_2)}{(\mu_x^2 + \mu_y^2 + C_1)(\sigma_x^2 + \sigma_y^2 + C_2)}, \tag{19}$$

where $\mu$, $\sigma^2$, and $\sigma_{xy}$ are local (Gaussian-windowed) statistics; $C_1, C_2$ stabilize division.

While these methods improved the evaluation of structural integrity, they still couldn't fully capture the complex textures and nuanced details that make an image look realistic. This limitation paved the way for learned perceptual metrics. The Learned Perceptual Image Patch Similarity (LPIPS) Zhang et al. (2018b) metric was a breakthrough, using the internal representations of a deep neural network (like VGG) to measure similarity in a way that closely mimics human judgment. Other metrics also target key visual features; FSIM (Feature Similarity Index) Zhang et al. (2011) focuses on phase congruency and gradient magnitude, which are critical to how we perceive edges and shapes. Pushing this concept further, DISTS (Deep Image Structure and Texture Similarity) Ding et al. (2020) uses a purpose-trained network to expertly balance the importance of structural correctness and textural realism, providing one of the most comprehensive full-reference evaluations available today.

**LPIPS**  It compares deep features from a fixed backbone (e.g., VGG). For layer $l$ with unit-normalized features $\hat{\phi}_l(\cdot)$ and learned channel weights $w_l$,

$$d_{\text{LPIPS}}(x,y) = \sum_l \frac{1}{H_l W_l} \sum_{h,w} \big\| \, w_l \odot \big(\hat{\phi}_l^{h,w}(x) - \hat{\phi}_l^{h,w}(y)\big)\big\|_2^2. \tag{20}$$

**FSIM**  It weights per-pixel similarity by phase congruency (PC) and gradient magnitude (GM). With

$$S_{\text{PC}} = \frac{2\,\text{PC}_x\,\text{PC}_y + T_1}{\text{PC}_x^2 + \text{PC}_y^2 + T_1}, \quad S_{\text{G}} = \frac{2\,\text{GM}_x\,\text{GM}_y + T_2}{\text{GM}_x^2 + \text{GM}_y^2 + T_2}, \tag{21}$$

the local similarity is $S_L = S_{\text{PC}}^{\alpha} S_{\text{G}}^{\beta}$, and the image-level score is

$$\text{FSIM}(x,y) = \frac{\sum_p \big[\max\{\text{PC}_x(p), \text{PC}_y(p)\}\, S_L(p)\big]}{\sum_p \max\{\text{PC}_x(p), \text{PC}_y(p)\}}. \tag{22}$$

**DISTS**  It decomposes similarity in deep space into structure and texture terms per layer $l$ of a fixed CNN:

$$\text{DISTS}(x,y) = \sum_l \alpha_l\big(1 - \rho(\phi_l(x), \phi_l(y))\big) \; + \; \beta_l\big(1 - \rho(\mu(\phi_l(x)), \mu(\phi_l(y)))\big), \tag{23}$$

where $\rho$ is correlation and $\mu(\cdot)$ denotes channel-wise means (texture statistics).

A significant challenge with all these metrics is their reliance on a perfect, HR ground truth image, which is often unavailable in real-world scenarios. This necessitates the use of no-reference, or blind, quality assessors. A classic example is NIQE (Natural Image Quality Evaluator) Mittal et al. (2012), which doesn't need a reference but instead measures how an image's statistical properties deviate from those of an ideal natural image. While NIQE provides a general sense of realism, modern approaches leverage complex deep learning for more accurate blind assessments. For instance MANIQA (Multi-dimension Attention Network for IQA) Yang et al. (2022) use sophisticated Transformer architectures to analyze images and predict a quality score that strongly aligns with human opinion.

**NIQE** fits a multivariate Gaussian (MVG) to natural-scene-statistics (NSS) features from pristine images, and another MVG to features from $x$; quality is the (Mahalanobis-type) distance between these Gaussians:

$$\text{NIQE}(x) = \sqrt{(\mu_n - \mu_x)^\top \left(\tfrac{\Sigma_n + \Sigma_x}{2}\right)^{-1} (\mu_n - \mu_x)} \quad \text{(lower is better).} \tag{24}$$

**MANIQA** uses a ViT backbone with transposed (channel) and Swin-based (spatial) attention blocks and a patch-weighted head to predict a scalar quality:

$$\hat{q} = f_\theta(x). \tag{25}$$

Finally, evaluating a generative SR model isn't just about the quality of a single image but also about the model's ability to produce a diverse and realistic distribution of outputs. The standard for this is the Fréchet Inception Distance (FID) Heusel et al. (2017), which measures the statistical difference between the feature distributions of many generated images and real images, effectively scoring both quality and variety.

**(Fréchet Inception Distance (FID)** Given Inception features for a set of SR images with $(\mu_g, \Sigma_g)$ and reference HR images with $(\mu_r, \Sigma_r)$, model each set as a Gaussian and compute the Fréchet (Wasserstein-2) distance:

$$\text{FID} = \|\mu_r - \mu_g\|_2^2 + \text{Tr}\Big(\Sigma_r + \Sigma_g - 2\left(\Sigma_r \Sigma_g\right)^{1/2}\Big). \tag{26}$$

We include five complementary criteria that target aspects of quality not fully captured by PSNR/SSIM. **SAM** quantifies spectral fidelity by measuring the angle (in degrees) between predicted and reference spectra, thus being insensitive to scale and directly penalizing spectral shape errors—crucial for multi/hyperspectral and color-constancy–sensitive tasks. $\mathbf{\Delta E_{00}}$ (CIEDE2000) measures perceptual color difference in CIELAB with SOTA corrections for lightness, chroma, and hue interactions; it aligns with human judgments and reveals small but perceptually important color shifts that PSNR can miss. **Gaussian NLL** evaluates probabilistic predictions by scoring the entire predicted distribution, rewarding sharp, accurate means and penalizing both bias and misestimated uncertainty ($\sigma$); lower NLL means better calibrated, better-fit likelihoods. Complementing this, $\textbf{ECE}_{\textbf{reg}}$ via quantile coverage assesses *calibration* of uncertainty: across target quantile levels, predicted quantiles should cover the empirical outcomes at the stated rates; deviations indicate over/under-confidence even when point accuracy is high. Finally, for pansharpening and related fusion, **QNR** jointly measures spectral consistency across bands ($D_\lambda$) and spatial detail preservation relative to the panchromatic guide ($D_s$), yielding a no-reference score that balances "do no spectral harm" with "add the right spatial detail." Alparone et al. (2007); Kuleshov et al. (2018); Sharma et al. (2005)

**Spectral Angle Mapper (SAM, in degrees).** For two spectra $\mathbf{x}, \mathbf{y} \in \mathbb{R}^B$,

$$\text{SAM}(\mathbf{x}, \mathbf{y}) = \frac{180}{\pi} \arccos\left(\frac{\mathbf{x}^\top \mathbf{y}}{\|\mathbf{x}\|_2 \|\mathbf{y}\|_2}\right). \tag{27}$$

Kruse et al. (1993)

$\Delta E_{00}$ **(CIEDE2000 color difference).** Given two CIELAB colors $(L_1^*, a_1^*, b_1^*)$ and $(L_2^*, a_2^*, b_2^*)$, define chroma $C_i^* = \sqrt{(a_i^*)^2 + (b_i^*)^2}$, mean $\bar{C}^* = \frac{1}{2}(C_1^* + C_2^*)$, and

$$G = \tfrac{1}{2}\left(1 - \sqrt{\frac{(\bar{C}^*)^7}{(\bar{C}^*)^7 + 25^7}}\right), \qquad a_i' = (1 + G)a_i^*, \quad C_i' = \sqrt{a_i'^2 + b_i^{*2}}, \quad h_i' = \text{atan2}(b_i^*, a_i'). \tag{28}$$

Let $\Delta L' = L_2^* - L_1^*$, $\Delta C' = C_2' - C_1'$, and

$$\Delta h' = \begin{cases} h_2' - h_1' & \text{if } |h_2' - h_1'| \le 180°, \\ h_2' - h_1' - 360° & \text{if } h_2' - h_1' > 180°, \\ h_2' - h_1' + 360° & \text{if } h_2' - h_1' < -180°, \end{cases} \qquad \Delta H' = 2\sqrt{C_1' C_2'} \, \sin\left(\frac{\Delta h'}{2}\right). \quad (29)$$

With means $\bar{L}' = \frac{1}{2}(L_1^* + L_2^*)$, $\bar{C}' = \frac{1}{2}(C_1' + C_2')$, and

$$\bar{h}' = \begin{cases} \frac{h_1' + h_2'}{2} & \text{if } |h_1' - h_2'| \le 180°, \\ \frac{h_1' + h_2' + 360°}{2} & \text{if } |h_1' - h_2'| > 180° \text{ and } h_1' + h_2' < 360°, \\ \frac{h_1' + h_2' - 360°}{2} & \text{otherwise,} \end{cases} \quad (30)$$

the weighting functions are

$$S_L = 1 + \frac{0.015(\bar{L}' - 50)^2}{\sqrt{20 + (\bar{L}' - 50)^2}}, \quad S_C = 1 + 0.045\,\bar{C}', \quad S_H = 1 + 0.015\,\bar{C}'\,T, \quad (31)$$

$$T = 1 - 0.17\cos(\bar{h}' - 30°) + 0.24\cos(2\bar{h}') + 0.32\cos(3\bar{h}' + 6°) - 0.20\cos(4\bar{h}' - 63°), \quad (32)$$

$$R_C = 2\sqrt{\frac{(\bar{C}')^7}{(\bar{C}')^7 + 25^7}}, \qquad R_T = -R_C \sin(2\Delta\theta), \quad \Delta\theta = 30° \exp\left[-\left(\frac{\bar{h}' - 275°}{25°}\right)^2\right]. \quad (33)$$

Finally, for parametric factors $k_L = k_C = k_H = 1$ (unless otherwise stated),

$$\Delta E_{00} = \sqrt{\left(\frac{\Delta L'}{k_L S_L}\right)^2 + \left(\frac{\Delta C'}{k_C S_C}\right)^2 + \left(\frac{\Delta H'}{k_H S_H}\right)^2 + R_T \, \frac{\Delta C'}{k_C S_C} \, \frac{\Delta H'}{k_H S_H}}. \quad (34)$$

Sharma et al. (2005)

**Gaussian Negative Log-Likelihood (per sample).** For targets $y_i$ and Gaussian predictions $\mathcal{N}(\mu_i, \sigma_i^2)$,

$$\text{NLL} = \frac{1}{2} \sum_{i=1}^{n} \left[\log(2\pi\sigma_i^2) + \frac{(y_i - \mu_i)^2}{\sigma_i^2}\right]. \quad (35)$$

**Expected Calibration Error (ECE) for regression via quantile coverage.** Let $\{\alpha_m\}_{m=1}^{M} \subset (0, 1)$ be nominal quantile levels and $\hat{q}_{\alpha_m}(x)$ the model's predicted $\alpha_m$–quantile for input $x$. Define the empirical coverage at level $\alpha_m$ by

$$\widehat{\text{cov}}(\alpha_m) = \frac{1}{n} \sum_{i=1}^{n} \mathbf{1}\{y_i \le \hat{q}_{\alpha_m}(x_i)\}. \quad (36)$$

Then an ECE-style scalar summary is

$$\text{ECE}_{\text{reg}} = \sum_{m=1}^{M} w_m \left|\widehat{\text{cov}}(\alpha_m) - \alpha_m\right|, \qquad w_m \ge 0, \ \sum_{m=1}^{M} w_m = 1, \quad (37)$$

(e.g., $w_m = \frac{1}{M}$). Lower is better; 0 indicates calibrated quantiles. Kuleshov et al. (2018)

**QNR (Quality with No Reference) for pansharpening, and its components.** Given the fused multispectral image $\hat{\mathbf{X}} = \{\hat{X}_b\}_{b=1}^B$, the original multispectral image $\mathbf{X} = \{X_b\}_{b=1}^B$ (upsampled to the fused resolution), and the panchromatic image $P$, define Wang–Bovik's universal image quality index $Q(\cdot, \cdot)$ applied bandwise (and to gradient images for spatial terms). Spectral distortion:

$$D_\lambda \;=\; \frac{2}{B(B-1)} \sum_{1 \le i < j \le B} \big| Q(X_i, X_j) - Q(\hat{X}_i, \hat{X}_j) \big|. \tag{38}$$

Spatial distortion:

$$D_s \;=\; \frac{1}{B} \sum_{b=1}^B \big| Q(\nabla X_b, \nabla P) \;-\; Q(\nabla \hat{X}_b, \nabla P) \big|. \tag{39}$$

With exponents $\alpha, \beta > 0$ (often $\alpha = \beta = 1$),

$$\mathrm{QNR} \;=\; (1 - D_\lambda)^\alpha \, (1 - D_s)^\beta, \quad \text{higher is better.} \tag{40}$$

Alparone et al. (2007)

**Implementation references** We rely on widely used implementations: LPIPS (official PyTorch), DISTS (official), MANIQA (official).[1]

### .3 PROMPTING DETAILS FOR CAPTION GENERATION

This section details the complete system prompt provided to the Vision-Language Model (GPT-5) (OpenAI, 2025) to generate descriptive captions from the HR reference images. The objective was to create semantically rich, resolution-aware descriptions of RTS while adhering to a natural, non-technical style suitable for guiding the SR process.

The prompt established an expert persona for the VLM and outlined a series of rules and constraints organized into style, content, and exclusions.

---

[1] https://github.com/richzhang/PerceptualSimilarity (LPIPS), https://github.com/dingkeyan93/DISTS (DISTS), https://github.com/IIGROUP/MANIQA (MANIQA).

---

**System Prompt Fed to the VLM**

**Persona & Objective**
You are an expert satellite image analyst writing captions. Your task is to describe images of permafrost thaw slumps. The goal is to create natural, descriptive captions suitable for training a text-to-image AI model. The captions must sound like a human describing a photo in simple terms.

**Style Rules**

- Use simple, everyday language.
- Write in a natural, fluid style.
- Use the present tense.
- Do not refer to "this image" or "the photo".

**Content Requirements**

- **Main Feature:** Describe the thaw slump using common terms like "landslide," "thaw slump," "ground collapse," or "erosion scar."
- **Shape & Form:** Mention its shape with simple descriptions like "crescent-shaped," "bowl-shaped," or "tongue of dirt."
- **Colors & Textures:** Describe the colors and textures of the ground, vegetation, and water (e.g., "dark brown soil," "green tundra," "cracked earth," "blue-green ocean").
- **Setting:** Briefly describe the surrounding environment, such as "coastal cliff," "green hillside," "tundra plain," or "riverbank."

**Exclusion Criteria (Crucial Constraints)**

- **NO JARGON:** Do not use technical terms like "RTS," "headwall," "lobe," or "rilled."
- **NO MEASUREMENTS:** Do not mention relative size, scale, or proportions (e.g., "covers a tenth of the scene," "a small feature").
- **NO ANNOTATIONS:** Do not mention map overlays, red lines, or other non-terrain elements.

---

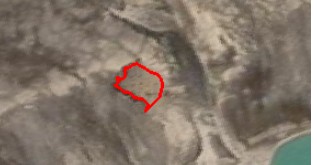

Figure 6: Example of input image given to VLM to generate an RTS-aware caption.

An example of a caption generated for the given input image 6:

"A bowl-shaped thaw slump cuts into a rugged coastal hillside, exposing dark brown soil and crumbly earth, with pale sandy streaks sliding downslope toward blue-green water, surrounded by gray-brown, sparsely vegetated tundra."

### .4 VALIDATION ON STANDARD SR

To rigorously assess the architectural robustness and general applicability of our proposed model, we conducted an additional set of experiments outside the primary cross-sensor domain. The objective of this analysis was to determine if the strong performance of our model is specific to the cross-sensor challenge or if its underlying design principles are fundamentally effective for the general task of Single Image Super-Resolution (SISR).

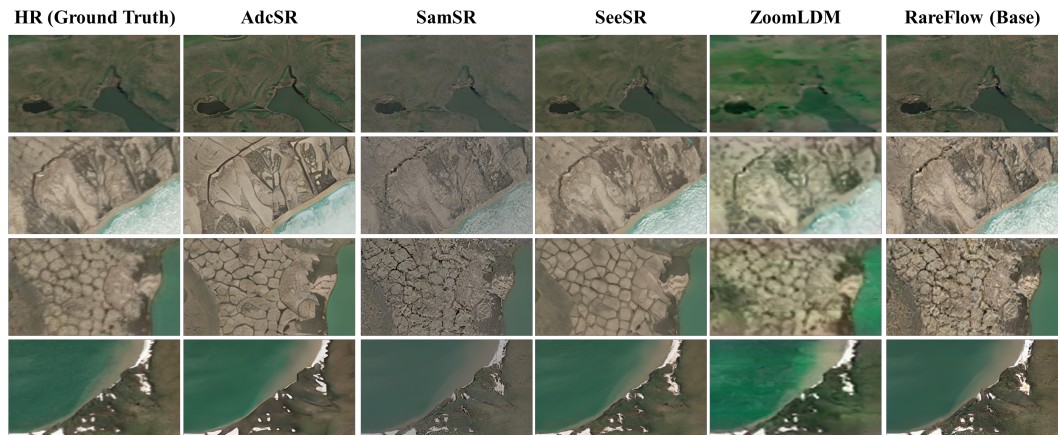

HR (Ground Truth)    AdcSR    SamSR    SeeSR    ZoomLDM    RareFlow (Base)

Figure 7: Qualitative comparison on HR-downsampled to HR data.

### .4.1 EXPERIMENTAL SETUP

We used the same HR ground truth images used in our main evaluation. For this SISR task, the corresponding LR inputs were synthesized by bicubically downsampling the HR images by a factor of 4x. The visual results in Figure 7 corroborate the quantitative findings. **RareFlow** excels at reconstructing fine-grained textures and sharp, plausible details that are often lost or blurred by competing methods. While other models tend to produce overly smoothed results, our model generates clean, realistic, and highly detailed images that are perceptually more convincing.

### .5 DOMAIN EXPERT EVALUATION

The primary objective of this evaluation was to rigorously assess our method across three key criteria: (1) the perceptual parity of our SR images with HR ground truth, (2) the clarity enhancement over LR inputs, and (3) the quality of the VLM-generated semantic guidance.

The evaluation was performed by a panel of domain experts to ensure the findings are grounded in practical scientific application. All participants are scientists actively conducting research on permafrost geomorphology and Arctic remote sensing at well-known international research centers. The cohort was composed of:

- **Research Scientists**, all holding a Ph.D. in a relevant field.
- **Senior Research Assistants**, with extensive, specialized experience in analyzing thaw slump features from satellite imagery.

The evaluation was conducted via a custom web-based interface (Figure 8). For each of the **30 samples** assigned to a reviewer, the interface displayed a comprehensive view containing four images: the HR ground truth with an RTS mask, the unmasked HR ground truth, our generated SR output, and the original Sentinel-2 LR input. The VLM-generated caption was displayed prominently alongside. This setup, while time-intensive for the experts, allowed for a thorough and direct comparison of all relevant data.

### .6 USE OF LARGE LANGUAGE MODELS

Large language models were utilized for grammatical correction, LaTeX formatting, debugging, and finding related work.

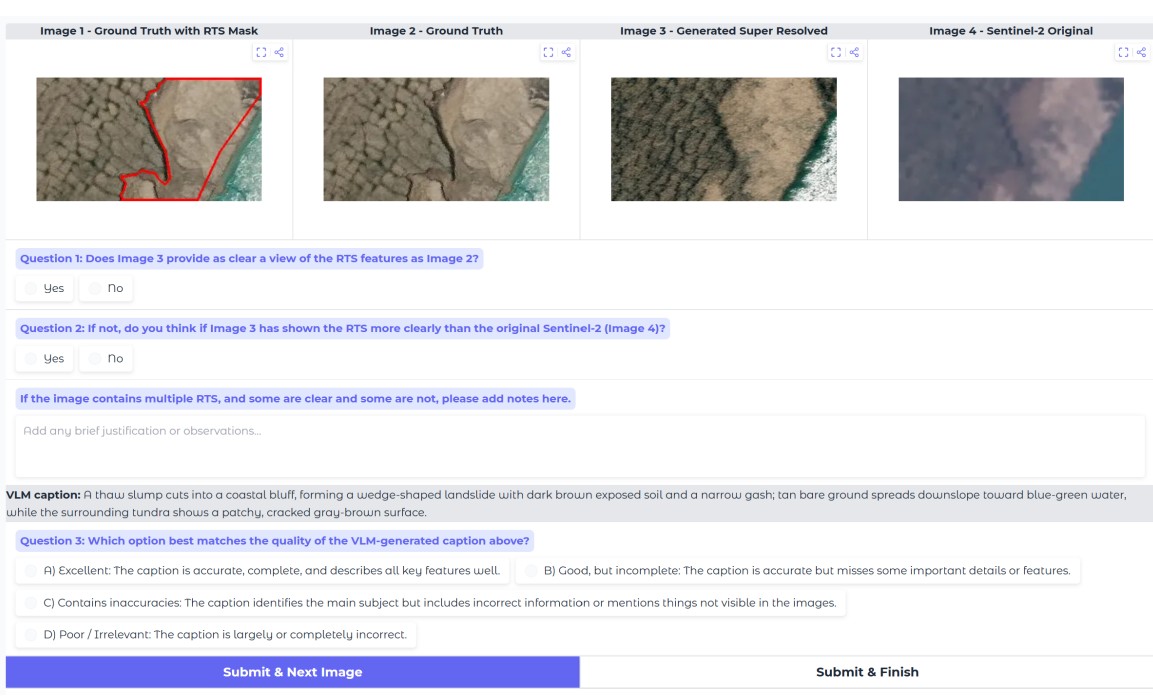

Figure 8: Our custom web-based interface for human evaluation.

