# OpenReview forum: "RareFlow: Physics-Aware Flow-Matching for Cross-Sensor Super-Resolution of Rare-Earth Features"
_ICLR.cc/2026/Conference — ICLR 2026 Conference Withdrawn Submission_

### Official Review · Reviewer_W1wK · 2025-10-20

**Soundness:** 2
**Presentation:** 2
**Contribution:** 2
**Rating:** 2
**Confidence:** 4

**Summary:**

This paper proposes a new single image super resolution approach tailored for a scientific context with low available data (less than one thousand images).
It is based on a ControlNet approach that attempts to propose a good balance the perception-distortion trade-off by using additional prompt information.
The prompts are obtained from GPT5.

**Strengths:**

The paper proposes a pipeline for training a ContolNet for super resolution.
One contribution is to modulate the control adapters using uncertainty.
The uncertainty is computed using Monte Carlo Dropout.
The remaining of the training pipeline is rather classical with a combined loss that assemble various desired properties (spectral faithfulness, color consistency, LPIPS, see Eq. (12)).

The reported experimental results are very good.
The method coined RareFlow achieves best performances on fidelity metrics, perceptual similarity and realism metrics (Table 1) for a difficult SISR task with intrument shift.
It is also very efficient in a more academic SISR problem with simulated LR images (no instrument shift).
A comprehensive ablation study is also performed.

**Weaknesses:**

The main weakness of this paper is the lack of novelty in its methodology for adapting a computer vision tool for a scientific application.
The use of GPT5 for the data expertise is also debatable as a scientific expertise.
All in all, even though this is an interesting problem, this may be of limited interest for the ICLR community.

Another main limitation is that reproducibility is not discussed.

Performance in terms of computation times are not discussed.
The computation of the uncertainty requires $T$ image evaluations at each step, which certainly burdens the computation time.

Splitting in training and testing set is not discussed.

If I understand correctly the GPT5 caption is performed on the HR reference image to perform SR on the LR image.
This is a form of ground truth leakage to provide this text to the network.

The paper suffers from several presentation issues:
* Some figures are not referenced in the text (Fig 2, Fig 5).
* ControlNet presentation: $f_\theta$ is presented as a frozen diffusion transformer, but then it appears as trained in Eq. (8). It would be good to highlight the dependency with respect to trainable parameters.
* It is not clear what are the latents $u_\theta$ and $u^\star$ from Eq. (9). There seem to be a conflict of notation with the uncertainty $u$ of Eq. (9).
* Caption of Fig. 2 discusses features $F_i$, a notation not present in the main text.
* What is the meaning of physics-aware? Prompt-informed?


Other minor issues:
* Appendix numbering,
* l. 048: RS used before definition line 054
* Figures referenced as Fig. and Figure
* 249: Eq. equation
* 246: equivalently: Does not sound equivalent
* Two references for ControlNet: Is it really needed to cite the v1 of the preprint in addition to the ICCV 2023 paper?

**Questions:**

1) About Eq. (9): Frequency regularization in latent space requires that the encoder $\mathcal{E}$ respects the low-mid-high frequency separation, but is has been shown that this is not generally the case in
Improving the Diffusability of Autoencoders, Skorokhodov et al., ICML 2025.
Is this ensured here ?

2) Figure 4: On the third line, can you ensure that snow is indeed present in the sentinel 2 LR or is it hallucination from training set?

3) Is the caption used for testing obtained from the groundtruth HR image ?

---

### Official Review · Reviewer_F295 · 2025-10-27

**Soundness:** 2
**Presentation:** 3
**Contribution:** 2
**Rating:** 2
**Confidence:** 4

**Summary:**

This paper proposes a new super-resolution framework for satellite images called RareFlow. It introduces three main challenges in realistic use-cases of super-resolution with satellite images: domain shift between sensors of which the data is combined, the need for preserving the original sensor properties in the images (which can be lost in super-resolving), and the presence of rare features that can be distorted during super-resolution. The authors additionally propose a new dataset to train and evaluate models for this problem. They present three evaluations: one real-world super-resolution evaluation with domain shift, using two different sensors (Sentinel-2 and Maxar); one with synthetic training pairs created by down-sampling Maxar data, and a qualitative evaluation by a human panel. The authors report improved performance over baseline super-resolution methods in both supervised and unsupervised metrics.

**Strengths:**

- The paper establishes a clear knowledge gap and highlights issues that are important to make super-resolution approaches usable in real-life satellite-image applications. Class imbalance/ lack of examples of rare phenomena are a pervasive problem that is important to address.
- The contributions are relevant, as the issues highlighted in the paper are specific to satellite images and are not necessarily a problem in the natural image domain (specifically the importance of preserving the ‘style’ of a sensor)

**Weaknesses:**

- Motivation/clarity can be improved: key methodological concepts (uncertainty quantification, physics-aware components, prompts, dual-conditioning, ControlNet) are introduced too late or not sufficiently motivated.
‘Uncertainty quantification’ is mentioned in the abstract, then the next occurrence is in section 3.2. It would be good to explain it also in the introduction and start of methods section.
The title mentions ‘physics-aware’, but this also does not appear in the introduction.
Prompts: Figure 1 shows that prompts are being used in RareFlow but it is not really clear from the introduction or the methods section how this fits into the methodology
‘dual-conditioning mechanism’ (line 085) – not yet clear what this is, can the authors elaborate?
ControlNet (line 086) can the authors elaborate why this architecture specifically is being used?

- Positioning/scoope: The paper positions the methods as a general method for satellite image datasets with rare features, yet the method is evaluated on a single small dataset with a single type of rare feature (retrogressive thaw slumps) and one combination of sensors, yet there is an enormous variety of possible rare features and sensor combinations. Therefore, the current evaluation does not support positioning this as a general method. Either the authors should narrow the scope of the paper, or present additional experiments on different datasets with other sensors and rare features.
- Experimental details: Critical experimental details are missing from the main text or relegated to appendices: training procedures (optimiser, epochs, random seed variations), evaluation metrics, and their justification. This limits the interpretation of the results and reproducibility. Furthermore, the dataset description lacks some details: the authors should clarify why the images are so small, why there are so few images in the dataset, and the issue of the bit depth (this issue is not unique to this dataset as satellite images tend to have higher bit-depth). This explanation should clarify why these issues have to be addressed algorithmically/with models rather than in pre-processing/dataset composition.
- Not all claims supported by evidence: The human expert validation (Section 4.3 and corresponding appendix lacks methodological rigour: number of panelists, number of images reviewed in total, panel demographics, inter-rater agreement, or quantitative results are reported. I suggest the authors provide more evaluation details as well as numerical results, or leave the analysis entirely out of the papers, as any conclusions should be supported by results.
- Conclusions: The conclusions should scope the contribution to remote sensing applications and acknowledge the single-dataset evaluation. The current text does not mention remote sensing data and is scoped too broadly.
- The image samples are too small to interpret. I suggest the authors update the figures to make the images larger (e.g. by removing some less relevant rows or columns to make space for larger images).
- Line 048: acronym RS not expanded
- Small formatting note: It seems like citations are placed after periods, which is slightly confusing (e.g. line 040, “European Space Agency (ESA) (a;b)”). Suggestion to add citations before periods and clearly distinguish them from the rest of the sentence.

**Questions:**

- ControlNet seems to be cited twice, in two different formats. Is this correct? (lines 738-743)
- Regarding the contributions: Contribution 2 mentions ‘we ensure the SR process preserves the large-scale radiometric information’ (lines 105-106), contribution 3 mentions ‘end-to-end model that jointly performs SR and radiometric harmonization’ (line 107). What is the difference between contributions 2 and 3, if they both refer to radiometric harmonisation?
- Lines 180-184: What domain-specific knowledge is necessary to reconstruct the RTS? What is the unique morphology of the RTS? These aspects should be made more concrete to explain why generic photo domain methods wouldn’t work.
- How representative are RTS of rare features in satellite images? What are examples of other features?

---

### Official Review · Reviewer_jpJK · 2025-11-02

**Soundness:** 3
**Presentation:** 3
**Contribution:** 3
**Rating:** 6
**Confidence:** 3

**Summary:**

This paper proposes RareFlow, a physics-aware dual-conditioning super-resolution framework designed for cross-sensor and out-of-distribution remote sensing imagery. The method integrates a dual-conditioning architecture combining Gated ControlNet and text-guided semantic conditioning, an uncertainty-gated control mechanism using Monte Carlo dropout and a physics-aware loss. Experiments demonstrate that RareFlow significantly improves perceptual realism and physical consistency over SOTA baselines such as SeeSR, AdcSR, and ZoomLDM.

**Strengths:**

The paper presents a well-motivated and technically innovative framework that effectively addresses the issue of generating physically inconsistent details in out-of-distribution scientific imagery. RareFlow’s dual-conditioning architecture successfully balances structural fidelity and semantic guidance, while its uncertainty-gated control mechanism adaptively suppresses hallucinated features under high uncertainty. The introduction of a physics-aware loss formulation, which integrates spectral alignment, radiometric consistency, and perceptual quality, provides a principled bridge between visual realism and physical correctness.

**Weaknesses:**

1.While the paper repeatedly emphasizes its “physics-aware”, the methodology relies primarily on heuristic loss formulations rather than an explicit physical modeling framework. No clear theoretical connection is established between the proposed losses and physics. As a result, the term “physics-aware” feels more empirical than principled, potentially overstating the scientific rigor of the approach.
2.Although most backbone components are frozen, the overall training pipeline still feels fragmented and under-specified, relying on several pretrained modules (VAE, SD3, ControlNet) whose interconnections are not fully transparent.
3.Key hyperparameters (e.g., λ-weights for each loss) are not justified or tuned systematically.

**Questions:**

See weaknesses.

---

### Official Review · Reviewer_U7Ez · 2025-11-02

**Soundness:** 2
**Presentation:** 3
**Contribution:** 3
**Rating:** 6
**Confidence:** 2

**Summary:**

This paper proposes RareFlow which is designed OOD-robust super-resolution in remote sensing, especially rare geomorphic features across heterogeneous sensors. The method combines flow-matching with dual conditioning: a Gated ControlNet to preserve fine-grained geometry from the LR input and text prompts to steer complex feature synthesis. Training uses physics-aware losses to enforce spectral/radiometric consistency with sensor properties, and a stochastic forward-pass uncertainty estimates unfamiliar inputs to mitigate hallucinations. A curated cross-sensor benchmark shows large qualitative gains and almost 40% FID reduction over SOTA baselines.

**Strengths:**

1) They designed losses tied to sensor spectra which helps keeping outputs physically plausible, not just visually sharp. This also ensure that the large-scale radiometric information are preserved and the hallucination is prevented.

2) A gated ControlNet (for structure) plus text prompts (for semantics) jointly handle appearance and geometry under distribution shift.

3) They got a strong performance on benchmark compared to SOTA methods.

**Weaknesses:**

1) How sensitive are the results to prompt wordings? Can you do some ablation on this?

2) Do the physics losses transfer to unseen sensors and bands without retraining? Can you provide any zero-shot evaluation?

3) Do the authors have any intuition why PSNR is low comparatively compared to other metrics in Table 3?

4)  Do the author have some ablations on different parts of the losses. Which components (spectral vs. radiometric) drive more gains? Provide loss term ablations.

**Questions:**

Check the weakness.

---

### Note · Authors · 2025-11-13

I have read and agree with the venue's withdrawal policy on behalf of myself and my co-authors.